# On Machine Learning in Clinical Interpretation of Retinal Diseases Using OCT Images

**DOI:** 10.3390/bioengineering10040407

**Published:** 2023-03-24

**Authors:** Prakash Kumar Karn, Waleed H. Abdulla

**Affiliations:** Department of Electrical, Computer and Software Engineering, University of Auckland, Auckland 1010, New Zealand

**Keywords:** OCT, fundus, machine learning, deep learning

## Abstract

Optical coherence tomography (OCT) is a noninvasive imaging technique that provides high-resolution cross-sectional retina images, enabling ophthalmologists to gather crucial information for diagnosing various retinal diseases. Despite its benefits, manual analysis of OCT images is time-consuming and heavily dependent on the personal experience of the analyst. This paper focuses on using machine learning to analyse OCT images in the clinical interpretation of retinal diseases. The complexity of understanding the biomarkers present in OCT images has been a challenge for many researchers, particularly those from nonclinical disciplines. This paper aims to provide an overview of the current state-of-the-art OCT image processing techniques, including image denoising and layer segmentation. It also highlights the potential of machine learning algorithms to automate the analysis of OCT images, reducing time consumption and improving diagnostic accuracy. Using machine learning in OCT image analysis can mitigate the limitations of manual analysis methods and provide a more reliable and objective approach to diagnosing retinal diseases. This paper will be of interest to ophthalmologists, researchers, and data scientists working in the field of retinal disease diagnosis and machine learning. By presenting the latest advancements in OCT image analysis using machine learning, this paper will contribute to the ongoing efforts to improve the diagnostic accuracy of retinal diseases.

## 1. Introduction

Before we start the targeted objectives of this paper, it is important to introduce some aspects in the eye’s biology and the role OCT in detecting common sicknesses. Figure 1 shows the eye structure and the impact of the macula in it function. The macula is a small 5 mm but essential part of the retina, located at the back of the eye, responsible for central vision, colour vision, and fine detail. It contains a high concentration of photoreceptor cells that detect light and send signals to the brain, which interprets them as images. The rest of the retina processes peripheral vision. There are several diseases associated with the macula, including diabetic macular oedema (DME), age-related macular degeneration (AMD), cystoid macular oedema (CME), and central serous chorioretinopathy (CSCR). “Oedema” refers to swelling, and swelling of the retina is termed macular oedema. The major causes of macular swelling include cyst formation, retinal fluid accumulation, and hard exudate formation due to prolonged diabetes [1]. A cross-sectional diagram of the eye is provided in Figure 1A. Age-related macular degeneration is a common macular disease seen in older adults over the age of 50 but can also occur at an early age [2]. The severity of this disease can be divided into early, intermediate, and late stages. The disease is primarily detected in the late stage when central vision begins to degrade, leading to blurred or complete vision loss. Early detection of AMD can be a preventive measure since there is no way to reverse the vision loss that has already occurred. The later stage is further classified as wet and dry AMD, with up to 90% of cases associated with dry AMD. Figure 1B compares the vision experienced by a healthy retina and a person suffering from macular degeneration.

### 1.1. Ophthalmic Imaging

Ophthalmic images help diagnose eye pathologies and monitor patients’ health progress. Some ophthalmic imaging techniques include fundus imaging, optical coherence tomography (OCT), fluorescein angiography, slit-lamp bio-micrography, and gonio-photography [3]. Each imaging technique has its application: fundus photo, which provides a 2D image of the retina; OCT, which gives the cross-section of the retina; fluorescein angiography, which shows the appearance of an active diseased cell; gonio-photograph, which provides a magnified and detailed view of the anterior cornea.

OCT is the most widely used imaging system to view the internal structure of tissues, nerves, retina, brain neurons, the molecular structure of the compound, and blood flow. It can effectively diagnose diseases such as diabetic retinopathy, glaucoma, age-related macular degeneration, macular oedema, central serous chorioretinopathy, solar retinopathy, cystoid macular oedema, and diabetic macular oedema [4]. As a result, OCT image analysis has become the focus of attention among researchers due to its potential to diagnose a wide range of retinal diseases.

### 1.2. OCT Types

OCT is an optical imaging modality that uses backscattered light to perform high-resolution, cross-sectional imaging of internal microstructures in materials and biological systems [5]. It measures the echo time delay and magnitude of the light to generate an image with a resolution of 1–15 μm, which is like ultrasound imaging but with better resolution [6]. OCT can also be defined as imaging by section using a light source with a constant phase difference. There are different types of OCT imaging techniques, as described below [7].

#### 1.2.1. Time-Domain OCT (TD-OCT)

This imaging system is based on a Michelson interferometer, which consists of a coherent light source, beam splitter, reference arm, and a photodetector. TD-OCT uses completely out-of-phase but with a constant phase difference as a light source with a manual adjustment of the reference arm.

#### 1.2.2. Fourier-Domain OCT (FD-OCT)

The working principle of this imaging technique is similar to TD-OCT, but it uses broadband light sources which are out of phase and have a variable phase difference. The light detected at the photodiode is a mixture of light with different wavelengths. These lights are passed through a spectrometer, and a Fourier analysis is carried out to decompose the signal into constituent elements. In this approach, the reference arm is fixed, and different retina layers are imaged with varying wavelength-based light sources. This imaging technique is also known as spectrometer-based OCT imaging or SD-OCT.

#### 1.2.3. Swept-Source OCT

This is also a Fourier-domain-based OCT imaging technique with a swept source as a light source. The optical setup is like FD-OCT, but the broadband light source is replaced by an optical source that rapidly sweeps a narrow linewidth over a broad range of wavelengths. Swept-source OCT uses a wavelength-sweeping laser and a dual-balanced photodetector, allowing faster A-scan with deeper penetration that enables the ophthalmologist to perform enhanced depth imaging (EDI). Various types of OCT images based on the image acquisition methodology and area are classified in Figure 2. To date, many ophthalmologists still rely on fundus imaging to diagnose eye diseases. However, despite its effectiveness, it has limitations. It cannot give the exact health condition of the macula (responsible for central vision), proper visualisation during severe retinal occlusion, etc. However, OCT imaging provides layer-wise detail of the retina, which helps ophthalmologists identify the exact cause of vision loss, the chances of vision recovery, the progress of treatment, and the effectiveness of various drugs. OCT comprises 11 layers and has a significant individual role in good vision. The retina is more likely to be affected by an intraretinal cyst, subretinal fluid, exudates, neovascularisation, and erosion of different layers [8]. The locations of these pathologies between different layers of OCT determine the disease. Hence, segmenting the OCT 11 layers is essential to detect those pathologies and classify the disorder using machine learning algorithms.

This paper is categorised into two parts. The first part deals with the anatomical interpretation of OCT images, followed by disease identification based on anatomical changes. The second part of this paper describes various machine-learning approaches for OCT image processing and disease identification. For medical image processing, it is crucial to learn the anatomical structure of an OCT image so that the problems faced by the ophthalmologist can be well addressed. Before developing any OCT image processing algorithm, one should know how doctors diagnose patients. Thus, this paper provides a detailed list of different characteristics, OCT terminologies, and clinical appearances of OCT images by which ophthalmologists can distinguish the health condition of the eyes.

## 2. Clinical Interpretation of OCT Images with Different Biomarkers

Clinical interpretation refers to the nomenclature used to understand the anatomy of OCT image, and the biomarker is the benchmark set by doctors to identify various retinal diseases. Before further explaining the terminologies used in OCT images, we introduce the human eye biology, retina, and macula.

The retinal macular OCT image consists of 11 different subretinal layers, as shown in Figure 3 [9]. These are the internal limiting membrane (ILM), retinal nerve fibre layer (RNFL), ganglion cell layer (GCL), inner plexiform layer (IPL), outer plexiform layer (OPL), outer nuclear layer (ONL), external limiting membrane (ELM), photoreceptor layer, retinal pigment epithelium (RPE), Bruch’s membrane (BM), and choroid layer. The red arrow shows the vitreous area, also called hyporeflective. The deep valley-like structure shown with a blue arrow is the fovea of the macula, which is responsible for central vision. The yellow arrow in Figure 3 represents the choriocapillaris or choroid region which is connected with the sclera.

The outer retinal layers are the most hyperreflective band composed of more than one line, termed the photoreceptor inner–outer segment junction and the RPE layer. The line highlighted in red in Figure 4 is the junction between the inner and outer segments of photoreceptors, whereas the band highlighted in green is the RPE. These lines often seem merged but are separate lines. A further RPE layer consists of two different layers believed to be RPE’s inner and outer membranes. Above the RPE and the photoreceptor inner-outer segment junction, the photoreceptor and the outer plexiform layer are present. The outer plexiform layer is spread with the entire length of RPE and the edge of the fovea. Another important layer of the retina is the retinal nerve fibre layer (RNFL), whose thickness determines the glaucoma disease’s presence. Sometimes RNFL is misinterpreted in the patient having posterior vitreous detachment (PVD).

PV is the outermost layer of the retina that prevents the retina from exposure to vitreous liquid. The green line in Figure 5 shows the RNFL, and the red line shows the posterior vitreous layer. The nasal side of the image can be identified from the thickness of the RNFL, where a thicker RNFL is seen towards the nasal area, and a thinner RNFL is seen towards the temporal region.

While interpreting the OCT image, we must know its associated artefacts. One of the artefacts is the shadow in the retina. A retinal blood vessel may shadow an OCT image during the image acquisition process that should not be interpreted as pathology. Other artefacts are vitreous floaters and arterial capillaries. Figure 6 shows some of the shadows in the OCT image.

The posterior face of the vitreous is often visible on the OCT scan. The information in these scans can assist in determining whether the PVD is complete or incomplete. In Figure 7, the posterior face of the vitreous is wholly detached from the temporal and nasal front of the retina, whereas it is attached to the fovea. Whenever the posterior vitreous is separated from the retina, it is called complete PVD or partial PVD. The distance between the retina and detached PV determines the stage of PVD.

Another pathology that is commonly seen in the retina is vitreomacular traction (VMT). The central macula densely supports the vitreous; during vitreous syneresis, the vitreous condenses and leads to PVD. However, if the posterior vitreous does not detach from the central macula, the vitreous will pull the macula, causing vitreomacular traction. Figure 7 is also an example of VMT, where the PV is still attached to the macula, and traction can be seen in the corresponding fundus image. VMT is also one of the main reasons for retinal thickening, resulting in retinal distortion and intraretinal cyst formation. Hence, detecting early PVD and VMT can be one of the early biomarkers for detecting cystoid macular oedema (CME). Some extreme results of VMT can be seen in Figure 8.

Figure 8A shows the VMT caused by partial PVD, leading to macular holes, subretinal fluid formation, etc. The red arrow mark in Figure 8B shows that the cystic space is open to the vitreous area, creating a lamellar hole with a roof of retina attached (highlighted in purple). In Figure 8C, the traction from the posterior face of the vitreous (highlighted in red) is extensively high, which results in the formation of cystic space within the retina (highlighted in orange) and subretinal fluid (highlighted in blue). The yellow line in the figure also shows that the junction of the photoreceptor inner and outer layer is separated by subretinal fluid. The graininess in Figure 8 is not due to low resolution but due to the patient suffering from a sclerotic nuclear cataract.

Another important biomarker for early macular oedema detection involves identifying an epiretinal membrane (ERM). An ERM is attached to the retina and is often as bright as a retinal nerve fibre layer (RNFL), making it challenging to identify. The shrinking of ERM can cause visual distortions (metamorphopsias) and blurred vision leading to retina traction. The effect of forming the epiretinal membrane and PVD is similar, resulting in macular oedema and macular degeneration.

Sometimes, it is challenging to distinguish ERM from RNFL in the OCT image, but clinical correlation can help distinguish between the two. In Figure 9, the inner nuclear layer (highlighted in green) has been pulled upwards with a peak (red arrow) towards the outer nuclear layer resulting from the formation of ERM highlighted in red. If the patient is left untreated at this stage, it may lead to a lamellar macular hole formation. This is the extreme effect of ERM, where the outer nuclear layer is exposed to the vitreous, and the patient’s central vision is compromised. A significant foveal depression shown in Figure 10 may look like a total thickness macular hole because of the upward wrinkle of the inner retinal layers. However, it should be noted that the hole does not go through retinal layers and seems to stop at the boundary of the outer nuclear layer. Deep depression in the fovea is often a lamellar macular hole rather than a full macular hole.

## 3. Most Common Disease Identification Using OCT Image Analysis

As mentioned in the earlier section, OCT images and the different biomarkers involved within them can be helpful in the early detection of the most commonly occurring eye diseases. The detected layers, cyst size, fluid accumulation volume, etc. can be used to classify the retinal image from healthy to unhealthy. In this section, we discuss various eye-related diseases which can be identified with the help of OCT image analysis.

### 3.1. Glaucoma

Glaucoma is a condition when the periphery vision of the person is gradually diminished towards the central vision and finally leads to point vision or technically permanent blindness [10]. This disease is the second leading cause of blindness in the world. There are different methods opted by the clinician to identify the glaucomatous eye, such as intraocular pressure (IOP), central corneal thickness (CCT), Humphrey visual field analysis, and optic cup-to-disc ratio [11]. The Euclidean distance between RNFL and ILM can also be an essential biomarker for detecting glaucoma. A study found that choroid thickness and the distance between Bruch’s membrane (BM) can be important biomarkers for glaucoma detection [12]. Recently, glaucoma has been detected by OCT image analysis only, which mostly relies on the classification of the RNFL thickness and minimum rim width (MRW) in different zones of the retina. A report from Nepal Eye Hospital for glaucoma detection using OCT image analysis is shown in Figure 11.

Optic nerve damage caused by glaucoma is typically characterised by thinning of the retinal nerve fibre layer (RNFL) and a decrease in the minimum rim width (MRW). To assess this damage, the eye is divided into six zones: inferior, superior, nasal, temporal, superior temporal, and superior nasal [13]. The distribution of RNFL thickness across these zones can help doctors determine the severity of vision loss. OCT (optical coherence tomography) is a useful tool for measuring progressive changes in RNFL thickness and MRW, and it can provide precise data on the eye’s anatomical composition. In addition to clinical examination, patient history, and visual field results, OCT can provide valuable information in the diagnostic process of glaucoma. An example of RNFL and MRW classification is shown in Figure 11, which illustrates a patient’s RNFL thickness and MRW distribution across various vision zones. The green line and shadow represent the thickness distribution benchmark taken from a European descent database collected in 2014, the yellow shadow indicates the thickness within the borderline limit and the red shadow indicates the thickness outside the standard boundary. The black line in Figure 11 represents the thickness distribution of the patient being evaluated, which is compared to the benchmark represented by the green line. The pie chart in Figure 11 displays the numerical values of RNFL thickness, including the amount of thinning in each zone of the eye.

Juan Xu et al. [14] proposed a software-based algorithm which can segment four different layers of the retina and compare them with the conventional circumpapillary nerve fibre layer (cpNFL) for the detection of glaucoma. Four different layers detected in this research were defined as macular nerve fibre layer (mNFL), a combination of ganglion cell layer–inner plexiform layer–nuclear layer, outer plexiform layer, and a combination of outer nuclear layer–photoreceptor layer. They used 47 subjects which contained 23 that were normal and 24 with glaucoma. It was found from the experiment that the average retinal thickness in glaucoma patients was significantly greater than in regular patients. Varmeer et al. [15] worked on segmenting six different subretinal layers, which applies to normal and glaucoma eyes. They could generate the thickness map of multiple layers with an RMS error between 4 and 6 µm. Jakub et al. [16] presented a study investigating IOP, vessel density (VD), and retinal nerve fibre layer in a patient who had never been treated for glaucoma. It is known that IOP ≥ 22 mmHG is directly associated with a diminished field of vision, but this paper also gave the dependency of VD and RNFL for an eye having IOP ≤ 20 mmHG. This relation can be helpful for the detection of normal tension glaucoma.

### 3.2. Age-Related Macular Degeneration (ARMD)

Macular degeneration affects the central vision of the person rather than the peripheral vision. The macula of an eye can be eroded by various means, such as the occurrence of drusen, growth of abnormal leaky blood vessels, formation of cysts, accumulation of fluid, and ageing. When the physical appearance of the macula has any sort of physical deformities and causes rapid loss of central vision, it is known as macular degeneration; when it occurs in aged people, it is called age-related macular degeneration. A person suffering from this disease cannot be fully blind as the peripheral vision is kept. There are two types of macular degeneration: dry macular degeneration and wet macular degeneration. The vision of a person with this disease is shown in Figure 1B.

Khalid et al. [17] presented a fusion of fundus and OCT-based classification of ARMD. This study proposed an automated segmentation algorithm which can detect drusen in the fundus and OCT images. The proposed system tested over 100 patients, which consisted of 100 fundus images and 6800 B-scan images of OCT. The author presented an SVM-based classification of features for drusen identification, further verified with a corresponding fundus image. Arnt-Ole et al. [18] coined a deep learning-based algorithm, OptiNet, to identify AMD from SD-OCT images. OptiNet is the modification of classical deep learning architecture with an addition of different parallel architecture created from filter feature from each layer; it was trained on a dataset containing approximately 600 AMD cases. Drusen is the early biomarker for the detection of AMD; its detection and segmentation play a crucial role in stopping the progression to the next disease.

### 3.3. Macular Oedema

As discussed earlier, the swelling of the macula by any means can be called macular oedema. There are various types of macular oedema, such as diabetic macular oedema (DME), cystoid macular oedema (CME), and clinically significant macular oedema (CSME). As the name suggests, swelling of the macular in a diabetic patient with the formation of some hard and soft exudates is called DME. This disease is most common in patients in a proliferative stage where new leaky blood vessels are developed. This disease can also occur when a cyst is formed between subretinal layers termed CME. When it is formed in the eyes’ fovea region affecting the vision, it is called clinically significant macular oedema. Roychowdhury et al. [19] localised cysts in OCT images with an identification of six different layers using a high-pass filter. The algorithm proposed by the author can segment the boundaries of a cyst; hence, it was able to define a cystoid area. In another article, the author produced a 3D thickness map of the image of a patient with DME. Some pre-processing followed by segmentation was used to extract six different retina layers. They also found that the thickness of the DME retinal layer was 1.5 times greater than that of healthy retinal layers. The association of the biomarkers with different eye sicknesses is given in Table 1.

### 3.4. Diabetic Retinopathy

Diabetic retinopathy (DR) is a leading cause of blindness among working-age adults and is caused by damage to the blood vessels in the retina due to diabetes. Early detection and management of DR are crucial in preventing vision loss, and optical coherence tomography (OCT) is a noninvasive imaging technique that is widely used for the detection and management of DR. OCT images provide detailed information about the retinal structure, which can be used to identify the presence and severity of DR.

Recently, machine learning (ML) and deep learning (DL) approaches have been applied to analyse OCT images for the diagnosis of DR. These methods have shown promise for the automated diagnosis of DR with high accuracy and performance compared to traditional methods.

One of the most widely used ML approaches for diagnosing DR from OCT images is the decision tree-based method. A decision tree is a type of ML algorithm that can be used for both classification and regression tasks. In a 2019 study [20], the authors proposed a decision tree-based method for diagnosing DR from OCT images. This paper proposed an automatic system for diabetic retinopathy detection from colour fundus images. It uses a combination of segmentation methods such as the Hessian matrix, ISODATA algorithm, and active contour to extract geometric features of blood vessels, and the decision tree CART algorithm is used to classify images into normal or DR. It achieved high accuracy of 96% in blood vessel segmentation and 93% in diabetic retinopathy classification.

Another popular ML approach for DR diagnosis from OCT images is the k-nearest neighbour (k-NN). k-NN is a nonparametric method that can be used for both classification and regression tasks. Bilal et al. [21] proposed a novel and hybrid approach for prior detection and classification of diabetic retinopathy (DR), using a combination of multiple models to increase the robustness of the detection process. The proposed method uses pre-processing, feature extraction, and classification steps using support vector machine (SVM), k-nearest neighbour (KNN), and binary trees (BT). It achieved a high accuracy of 98.06%, a sensitivity of 83.67%, and 100% specificity when tested on multiple severities of disease grading databases.

Recently, DL approaches, specifically convolutional neural networks (CNNs), have been widely used to diagnose DR from OCT images. CNNs are a type of deep learning architecture that is particularly well-suited for image analysis tasks, such as image classification and segmentation.

One example of the diagnosis of DR from OCT images was presented in a research article [22] published in 2022, in which the authors proposed a three-step system for diabetic retinopathy (DR) detection utilising optical coherence tomography (OCT) images. The system segments retinal layers, extracts 3D features, and uses backpropagation neural networks for classification. The proposed system has an accuracy of 96.81%, which is an improvement over related methods using a leave-one-subject-out (LOSO) cross-validation.

More recently, the attention mechanism has been added to CNNs to improve the performance of the model by selectively highlighting the important features of the image. Sapna et al. [23] proposed an attention-based deep convolutional neural network (CNN) model to classify common macular diseases using scans from optical coherence tomography (OCT) imaging. The proposed architecture uses a pretrained model with transfer learning and a deformation-aware attention mechanism to encode variations in retinal layers, without requiring any pre-processing steps. It achieves state-of-the-art performance and better diagnostic performance with a reduced number of parameters.

It is worth noting that the implementation and customisation of the architectures, as mentioned above, are vital to achieving good performance, and hyperparameter tuning plays a crucial role. Additionally, many studies used pre-processing techniques such as image enhancement and image segmentation to enhance the features of the images that can be used for classification.

Another important aspect is using a large and representative dataset for training the model—datasets used for the studies cited above range from a few hundred to thousands of images. Having a large and diverse dataset can significantly improve the model’s performance.

It is also worth mentioning that these methods, although they have shown good performance, are not meant to replace the traditional diagnostic methods; rather, they should be used as a support tool to aid in the diagnosis and management of diabetic retinopathy. It is important to note that these algorithms are only as good as the data they were trained on, and there may be some cases where the model performs poorly. Therefore, it is essential to have a trained professional to interpret the results and make the final diagnosis.

Another important aspect to consider is the interpretability of these models. While traditional diagnostic methods have a clear and transparent decision-making process, deep learning models are known for their black-box nature, making it difficult to understand the reasoning behind the model’s predictions. This lack of interpretability can hinder the adoption of these methods in the clinical setting. Therefore, there has been a recent focus on developing interpretable models and methods for understanding the decision-making process of deep learning models in medical imaging.

In conclusion, deep learning-based methods have shown great potential in the classification of diabetic retinopathy from OCT images, achieving high accuracy and performance compared to traditional methods.

## 4. Analysis of Optical Coherence Tomography Images

This section focuses on analysing the retinal OCT images using various image processing algorithms and machine learning approaches. The images acquired from the patients in raw format might be associated with poor contrast, noise, low light, and other artefacts. Several image enhancement techniques are needed to extract useful information. Several approaches can be used to enhance the quality of the OCT image, such as denoising the images to classify diseases through pathologies segmentation.

### 4.1. Denoising OCT Images

Speckle noise is one of the most common noises generated in OCT imaging. Although imaging technology and equipment are continuously updated, speckle noise has not yet been fully solved. It degrades the performance of the automatic OCT image analysis [24]. Many hardware-based methods, which depend on specially designed acquisition systems, have been proposed for speckle noise suppression during imaging. Iftimia et al. [25] suggested an angular compounding by path length encoding (ACPE) technique that performs B-scan faster to eliminate speckle noise in OCT images. Kennedy et al. [26] presented a speckle reduction technique for OCT based on strain compounding. Based on angular compounding, Cheng et al. [27] proposed a dual-beam angular compounding method to reduce speckle noise and improve the SNR of OCT images. However, all these techniques can be applied to commercial OCT devices, adding an economic burden and increasing scan costs. Many algorithms [28,29,30] have been proposed to obtain a high signal-to-noise ratio (SNR) and high-resolution B-scan images from low-profile retinal OCT images. Bin Qui et al. [31] proposed a semi-supervised learning approach named N2NSR-OCT to generate denoised and super-resolved OCT images simultaneously using up- and down-sampling networks (U-Net (Semi) and DBPN (Semi). Meng-wang et al. [32] used a semi-supervised model to denoise the oct image using a capsule conditional generative adversarial network (Caps-cGAN) with few parameters. Nilesh A. Kande et al. [33] developed a deep generative model, called SiameseGAN, for denoising images with a low signal-to-noise ratio (LSNR) equipped with a Siamese twin network. However, these approaches achieved high efficiency because they were tested over synthesised noisy images created by the generative adversarial network.

Another approach to denoise the OCT images was proposed in [34], which captures repeated B-scans from a unique position. Image registration and averaging are performed to denoise the images. Huazong Liu et al. [35] used dual-tree complex wavelet transform for denoising the OCT images. In this approach, the image is decomposed into wavelet domain using dual-tree complex wavelet transform, and then the signal and noise are separated on the basis of Bayesian posterior probability. An example of OCT image denoising via super-resolution reconstruction is given in Figure 12 [36]. They used a multi-frame fusion mechanism that merges multiple scans for the same scene and utilises sub-pixel movements to recover missing signals in one pixel, significantly improving the image quality.

Usually, all OCT image analysis methods proposed in the literature consist of a pre-processing step before performing any main processing steps. Table 2 shows a relatively complete classification of denoising algorithms employed in OCT segmentation. This table shows median and nonlinear anisotropic filters are the most popular methods in OCT image denoising. The fundamental problem associated with most denoising algorithms is their intrinsic consequence in decreasing the image resolution.

The popularity of methods, like nonlinear anisotropic filters and wavelet diffusion, can be attributed to their ability to preserve edge information. It is also essential to mention that many recently developed algorithms that utilise graph-based algorithms are independent of noise and do not use any denoising algorithm.

### 4.2. Segmentation of Subretinal Layers of OCT Images

Segmentation is the next step in image processing after image denoising. This procedure gives detailed information about the OCT image ranging from pathologies to the measurement of subretinal layer thickness. While numerous methods [37,40,43,46] for retinal layer segmentation have been proposed in the literature for human OCT images, Sandra Morales et al. [47] used local intensity profiles and fully convolution neural networks to segment rodent OCT images. Rodent OCT images have no macula or fovea, and their lenses are relatively larger. For a single patient, this consists of 80% rodent OCT from a single scan. This approach successfully segmented all the layers of OCT images for healthy patients; however, it could only detect five layers in a diseased image. Yazdanpanah et al. [38] proposed a semi-automated algorithm based on the Chan–Vese active contours without edges to address six-retinal-layer boundary segmentation. They applied their algorithm to 80 retinal OCT images of seven rats and achieved an average Dice similarity coefficient of 0.84 over all segmented retinal layers. Mishra et al. [44] used a two-step kernel-based optimisation to segment retinal layers on a set of OCT images from healthy and diseased rodent retinas. Examples of efficient segmentation and false segmentation of OCT layers are given in Figure 13 and Figure 14. These images were captured with a Heidelberg OCT device at Nepal Eye Hospital.

### 4.3. Detection of Various Pathologies in OCT Images

The detection of various pathologies is the main OCT image analysis objective. A semi-automated 2D method based on active contours was used as the basis for the initial work on fluid segmentation in OCT [48]. In this approach, to segment the IRF and SRF, users need to point out each lesion to get detected by active contour. Later, in [49], a comparable level set method with a quick split Bregman solver was employed to automatically create all candidate fluid areas, which were manually eliminated or selected. These methods are challenging, and few practitioners use them. A multiscale convolutional neural network (CNN) was first proposed for patch-based voxel classification in [50]. It could differentiate between IRF and SRF fluid in both supervised and weakly supervised settings. Although it is the most important aspect in treating eye disease, automated fluid presence detection has received significantly less attention. Table 3 provides a summary of the presented fluid segmentation algorithms, comparing them in terms of many characteristics. An example of fluid segmentation is shown in Figure 15. In this figure, intraretinal fluid (IRF) is represented in red, subretinal fluid (SRF) is depicted in green, and pigment epithelial detachment (PED) is shown in blue.

### 4.4. Deep Learning Approach for OCT Image Analysis

Recent developments in AI, including deep learning and machine learning algorithms, have enabled the analysis of OCT images with unprecedented accuracy and speed. These algorithms can segment OCT images, detect pathological features, and identify biomarkers of various ocular diseases. Additionally, the use of AI frameworks such as TensorFlow and Keras has made it easier to develop these algorithms. AI is used for a wide range of applications in OCT, including image segmentation, disease diagnosis, and progression monitoring. One significant application of AI is the detection of age-related macular degeneration (AMD), a leading cause of blindness in older adults. AI algorithms can detect early signs of AMD, which can lead to early intervention and better outcomes for patients. Additionally, AI is used for the diagnosis and monitoring of other ocular diseases, including glaucoma and diabetic retinopathy. The use of AI in OCT has several potential clinical implications. One significant benefit is increased accuracy in diagnosis and treatment. AI algorithms can analyse OCT images with high precision and speed, leading to faster and more accurate diagnosis of ocular diseases. Additionally, AI can help clinicians monitor disease progression and evaluate treatment efficacy. This can lead to better patient outcomes and improved quality of life.

However, there are also potential drawbacks to using AI in OCT. One potential concern is the need for high-quality datasets. To develop accurate and reliable AI algorithms, large datasets of OCT images are required. Additionally, there is a risk of over-reliance on AI, which could lead to a reduction in human expertise and clinical judgment.

Deep learning approaches, particularly convolutional neural networks (CNNs), have been widely used in recent years for the analysis of optical coherence tomography (OCT) images. The success of these methods is attributed to the ability of CNNs to automatically learn features from the data, which can improve the performance of image analysis tasks compared to traditional methods.

One of the most promising deep learning architectures for OCT image analysis is the convolutional neural network (CNN). CNNs are designed to process data with a grid-like topology, such as images, and they are particularly well suited for tasks such as image classification and segmentation.

U-Net architecture is one of the most popular CNN architectures used in OCT image analysis. U-Net is an encoder–decoder type of CNN designed for image segmentation tasks. The encoder portion of the network compresses the input image into a low-dimensional feature space. In contrast, the decoder portion upscales the feature maps to the original image resolution to produce the final segmentation mask. A method using two deep neural networks (U-Net and DexiNed) was proposed in [2] to segment the inner limiting membrane (ILM), retinal pigment epithelium, and Bruch’s membrane in OCT images of healthy and intermediate AMD patients, showing promising results, with an average absolute error of 0.49 pixel for ILM, 0.57 for RPE, and 0.66 for BM.

Another type of architecture that is being used in the analysis of OCT images is Residual Network (ResNet). ResNet is a type of CNN which deals with the problem of vanishing gradients by introducing the idea of “shortcut connections” in the architecture. The shortcut connections bypass the full layers of the network and allow the gradients to be directly propagated to the earlier layers. A study proposed in [59] used a lightweight, explainable convolutional neural network (CNN) architecture called DeepOCT for the identification of ME on optical coherence tomography (OCT) images, achieving high classification performance with 99.20% accuracy, 100% sensitivity, and 98.40% specificity. It visualises the most relevant regions and pathology on feature activation maps and has the advantage of a standardised analysis, lightweight architecture, and high performance for both large- and small-scale datasets.

Subretinal fluid is a common finding in OCT images and can be a sign of various ocular conditions such as age-related macular degeneration, retinal detachment, and choroidal neovascularisation [60]. Accurate and efficient segmentation of subretinal fluid in OCT images is an essential task in ophthalmology, as it can help diagnose and manage these conditions.

However, manual segmentation of subretinal fluid in OCT images is time-consuming, labour-intensive, and prone to observer variability [60]. To address this challenge, there has been growing interest in using automated approaches to segment subretinal fluid in OCT images. In recent years, deep learning models have emerged as a promising solution for this task due to their ability to learn complex features and patterns from large amounts of data [60].

Azade et al. [61] introduced Y-Net, an architecture combining spectral and spatial domain features, to improve retinal optical coherence tomography (OCT) image segmentation. This approach outperformed the U-Net model in terms of fluid segmentation performance. The removal of certain frequency ranges in the spectral domain also demonstrated the impact of these features on the outperformance of fluid segmentation. This study’s results demonstrated that using both spectral and spatial domain features significantly improved fluid segmentation performance and outperformed the well-known U-Net model by 13% on the fluid segmentation Dice score and 1.9% on the average Dice score.

Bilal Hassan et al. [62] proposed using an end-to-end deep learning-based network called RFS-Net for the segmentation and recognition of multiclass retinal fluids (IRF, SRF, and PED) in optical coherence tomography (OCT) images. The RFS-Net architecture was trained and validated using OCT scans from multiple vendors and achieved mean F1 scores of 0.762, 0.796, and 0.805 for the segmentation of IRF, SRF, and PED, respectively. The automated segmentation provided by RFS-Net has the potential to improve the efficiency and inter-observer agreement of anti-VEGF therapy for the treatment of retinal fluid lesions.

Kun Wang et al. [63] proposed the use of an iterative edge attention network (EANet) for medical image segmentation. EANet includes a dynamic scale-aware context module, an edge attention preservation module, and a multilevel pairwise regression module to address the challenges of diversity in scale and complex context in medical images. Experimental results showed that EANet outperformed state-of-the-art methods in four different medical segmentation tasks: lung nodule, COVID-19 infection, lung, and thyroid nodule segmentation.

Jyoti Prakash et al. [64] presented an automated algorithm for detecting macular oedema (ME) and segmenting cysts in optical coherence tomography (OCT) images. The algorithm consists of three steps: removal of speckle noise using edge-preserving modified guided image filtering, identification of the inner limiting membrane and retinal pigment epithelium layer using modified level set spatial fuzzy clustering, and detection of cystoid fluid in positive ME cases using modified Nick’s threshold and modified level set spatial fuzzy clustering applied to green channel OCT images. The algorithm was tested on three databases and showed high accuracy and F1-scores in detecting ME cases and identifying cysts. It was found to be efficient and comparable to current state-of-the-art methodologies.

Menglin Wu et al. [65] proposed a two-stage algorithm for accurately segmenting neurosensory retinal detachment (NRD)-associated subretinal fluid in spectral domain optical coherence tomography (SD-OCT) images. The algorithm is guided by Enface fundus imaging, and it utilises a thickness map to detect fluid abnormalities in the first stage and a fuzzy level set method with a spatial smoothness constraint in the second stage to segment the fluid in the SD-OCT scans. The method was tested on 31 retinal SD-OCT volumes with central serous chorioretinopathy and achieved high true positive volume fractions and positive predictive values for NRD regions and for discriminating NRD-associated subretinal fluid from subretinal pigment epithelium fluid associated with pigment epithelial detachment.

Zhuang Ai et al. [66] proposed a fusion network (FN)-based algorithm for the classification of retinal optical coherence tomography (OCT) images. The FN-OCT algorithm combines the InceptionV3, Inception-ResNet, and Xception deep learning algorithms with a convolutional block attention mechanism and three different fusion strategies to improve the adaptability and accuracy of traditional classification algorithms. The FN-OCT algorithm was tested on a dataset of 108,312 OCT images from 4686 patients and achieved an accuracy of 98.7% and an area under the curve of 99.1%. The algorithm also achieved an accuracy of 92% and an AUC of 94.5% on an external dataset for the classification of retinal OCT diseases.

In addition to these architectures, several other techniques such as Attention U-Net, DenseNet, and SE-Net have also been proposed for OCT image analysis. Attention U-Net is a specific type of U-Net architecture that uses attention mechanisms to enhance the performance of segmentation tasks. A study [67] conducted in 2022 proposed a deep learning method called the wavelet attention network (WATNet) for the segmentation of layered tissue in optical coherence tomography (OCT) images. The proposed method uses the discrete wavelet transform (DWT) to extract multispectral information and improve performance compared to other existing deep learning methods.

DenseNet is another CNN architecture that is widely used for image classification tasks. It is known for its high efficiency and ability to learn features from multiple layers. A 2020 study [68] proposed a method for early detection of glaucoma using densely connected neural networks, specifically a DenseNet with 121 layers pretrained on ImageNet. The results obtained on a dataset of early and advanced glaucoma images achieved high accuracy of 95.6% and an F1-score of 0.97%, indicating that CNNs have the potential to be a cost-effective screening tool for preventing blindness caused by glaucoma; the DenseNet-based approach outperformed traditional methods for glaucoma diagnosis.

SE-Net is another variant of CNN architecture that includes squeeze and excitation blocks to improve the model’s performance by selectively highlighting the important features of the image. Zailiang Chen et al. [69] proposed a deep learning method called SEUNet for segmenting fluid regions in OCT B-scan images, which is crucial for the early diagnosis of AMD. The proposed method is based on U-Net and integrates squeeze and excitation blocks; it is able to effectively segment fluid regions with an average IOU coefficient of 0.9035, Dice coefficient of 0.9421, precision of 0.9446, and recall of 0.9464.

Overall, it can be concluded that deep learning-based methods have shown great potential in the analysis of OCT images and have outperformed traditional methods in various tasks such as image segmentation and diagnosis of retinal diseases. Different CNN architectures, such as U-Net, ResNet, Inception, Attention U-Net, DenseNet, and SE-Net have been proposed and used for these tasks with good performance. However, it should be noted that the success of these models also depends on the quality of the dataset used for training, and the generalisability of these models to new datasets needs further research. Common methods for deep learning-based OCT image processing are shown in Figure 16.

## 5. Publicly Available OCT Dataset

In most research on OCT, images are collected locally and are not made public, limiting the research scope in OCT image processing. Among the datasets used in the literature, most are either unclassified or not publicly available. Some of the datasets which are available publicly are given below.

Retouch Dataset: This dataset was created during a challenge conducted by MICCAI 2017. It comprises 112 macula-centred OCT volumes, each comprising 128 B-scan images. The images are captured from three different vendors: Zeiss, Spectralis, and Topcon. Download link: https://retouch.grand-challenge.org/ (Accessed date 16 February 2023).

Duke Dataset: This is a publicly available dataset collected through the joint effort of Duke University, Harvard University, and the University of Michigan. This dataset consists of 3000 OCT images from 45 participants: 15 with dry age-related macular degeneration, 15 with diabetic macular oedema, and 15 healthy patients. Download link: http://people.duke.edu/~sf59/Srinivasan_BOE_2014_dataset.htm (Accessed date 16 February 2023).

Labelled OCT and Chest X-ray: This dataset consists of 68,000 OCT images. The images are split into a training set and a testing set of independent patients. Images are labelled as train tests, validated, and divided into four directories: CNV, DME, drusen, and normal. This is probably the most populated of available datasets. Download link: https://data.mendeley.com/datasets/rscbjbr9sj/3 (Accessed date 16 February 2023).

Annotated Retinal OCT Images database—AROI database: This is a restricted dataset only available for research upon request. This database consists of a total of 3200 B-scans collected from 25 patients. The image acquisition was made with the Zeiss Cirrus HD OCT 4000 device. Each OCT volume consisted of 128 B-scans with a resolution of 1024 × 512 pixels. All images are annotated with pathologies such as pigment epithelial detachment (PED), subretinal fluid and subretinal hyperreflective material (marked jointly as SRF), and intraretinal fluid (IRF). Download link: https://ipg.fer.hr/ipg/resources/oct_image_database (Accessed date 16 February 2023).

The University of Waterloo Database: This is a publicly available dataset comprising more than 500 high-resolution images categorised into different pathological conditions. The image classes include normal (NO), macular hole (MH), age-related macular degeneration (AMD), central serous retinopathy (CSR), and diabetic retinopathy (DR). These images were collected from a raster scan protocol with a 2 mm scan length and 512 × 1024 resolution. Download link: https://dataverse.scholarsportal.info/dataverse/OCTID (Accessed date 16 February 2023).

Optima Dataset: The OPTIMA dataset is part of the OPTIMA Cyst Segmentation Challenge of MICCAI 2015. This challenge segments IRF from OCT scans of vendors Cirrus, Spectralis, Nidek, and Topcon. The dataset consists of 15 scans for training, eight scans for stage 1 testing, seven scans for stage 2 testing, and pixel-level annotation masks provided by two different graders. Download link: https://optima.meduniwien.ac.at/optima-segmentation-challenge-1/ (Accessed date 16 February 2023).

University of Auckland Dataset: This is a private dataset collected from two eye hospitals of Nepal: Dristi Eyecare and Nepal Eye Hospital. It consists of volumetric scans from two vendors, Topcon and Spectralis, with resolutions of 320 × 992 and 1008 × 596, respectively. Images from Topcon consist of a volumetric scan of 270 patients, where each volumetric scan consists of 260 B-scans on average. Similarly, images captured from Spectralis consist of a volumetric scan of 112 patients, and each volumetric scan consists of 98 B-scans from both eyes. All patient details are removed and anonymised. The ophthalmologists collaborating on image acquisition and disease classification are Mr Ravi Shankar Chaudhary from Dristi Eyecare and Mrs Manju Yadav from Nepal Eye Hospital. Dr Neyaz (MD, retinal specialist) from Nepal Eye Hospital classified various pathologies as the first observer, later verified by Dr Bhairaja Shrestha (MD, ophthalmologist), acting as the second observer. This dataset is available upon request (w.abdulla@auckland.ac.nz) and will be made available for public use.

## 6. Conclusions

This article highlighted the significance of using contemporary machine learning techniques in analysing optical coherence tomography (OCT) images for diagnosing various retinal diseases. The review showcased and investigated the recent advancements in the field of OCT image analysis and the potential of deep learning algorithms to improve diagnostic accuracy. However, we also highlighted the limitations of current approaches, such as the need for clinical interpretation and the lack of focus on early-stage disease detection. For example, there is a similarity in brightness between ERM and RNFL, and the misinterpretation of lamellar and full macular holes may lead to the misdiagnosis of the disease in its early stage. Furthermore, early-stage PVD and ERM detection is currently lacking in the literature, despite being essential for predicting cystoid macular oedema. Therefore, this paper also highlighted the importance of biomarkers for the early detection of diseases, such as the hill-like structure similar to drusen caused by ERM, which is a significant biomarker for choroidal neovascularisation.

Our investigation suggests that future studies should incorporate biomarkers for early disease detection in their deep learning models and aim to develop more robust and generalisable models that can handle variations in imaging conditions and patient populations. Additionally, integrating other imaging modalities and data types, such as fundus photography or patient clinical history, can improve the accuracy of disease detection.

In summary, using machine learning in OCT image analysis has shown promising results in diagnosing retinal diseases. Continued research in this field will significantly advance early eye disease detection, diagnosis, and management.

## Figures and Tables

**Figure 1 bioengineering-10-00407-f001:**
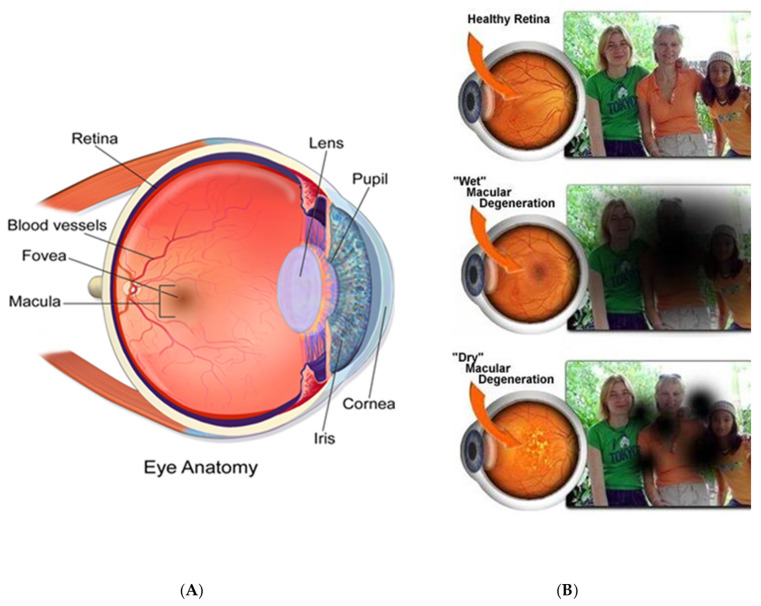
(**A**) Cross-section of eye; (**B**) vision after macular degeneration.

**Figure 2 bioengineering-10-00407-f002:**
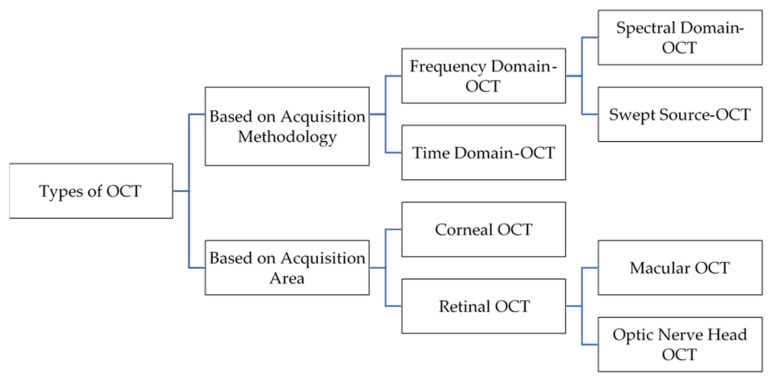
Classification of different types of OCT.

**Figure 3 bioengineering-10-00407-f003:**
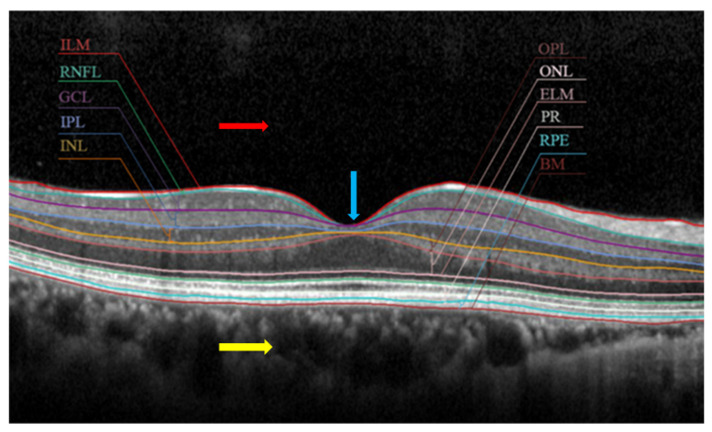
Different layers of macular OCT: vitreous (red), fovea (blue), and choroid (yellow).

**Figure 4 bioengineering-10-00407-f004:**
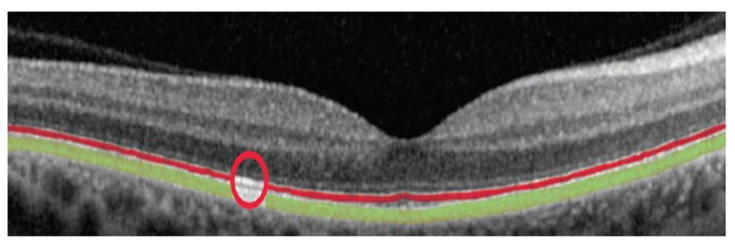
Outer retinal layer (red—photoreceptor, green—RPE, red circle—junction of photoreceptor).

**Figure 5 bioengineering-10-00407-f005:**
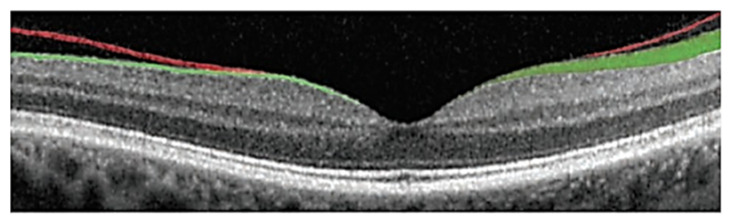
RNFL (green) and posterior vitreous layer (red).

**Figure 6 bioengineering-10-00407-f006:**
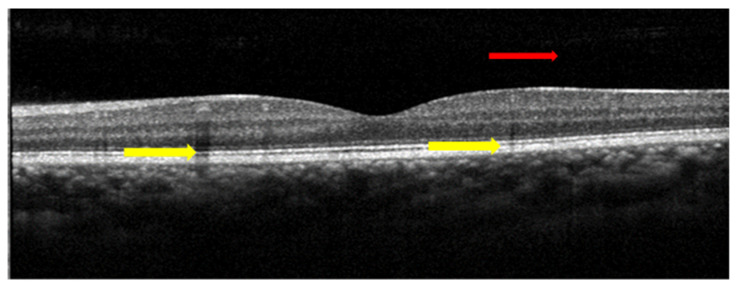
Shadows in OCT: blood vessels (yellow); floaters (red).

**Figure 7 bioengineering-10-00407-f007:**
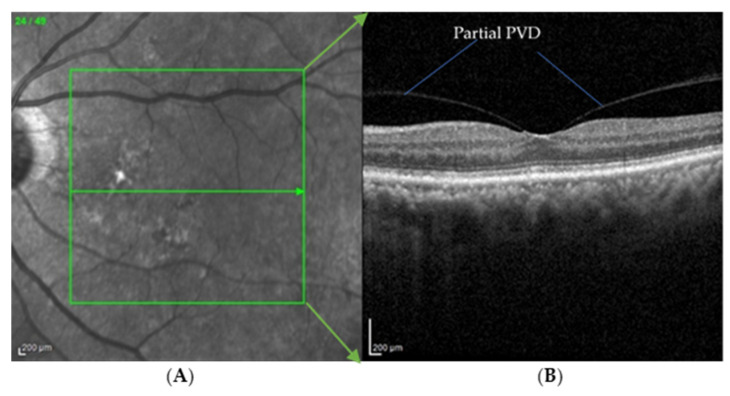
Partial PVD in OCT image: (**A**) fundus image; (**B**) corresponding OCT Image below the green arrow inside the box.

**Figure 8 bioengineering-10-00407-f008:**
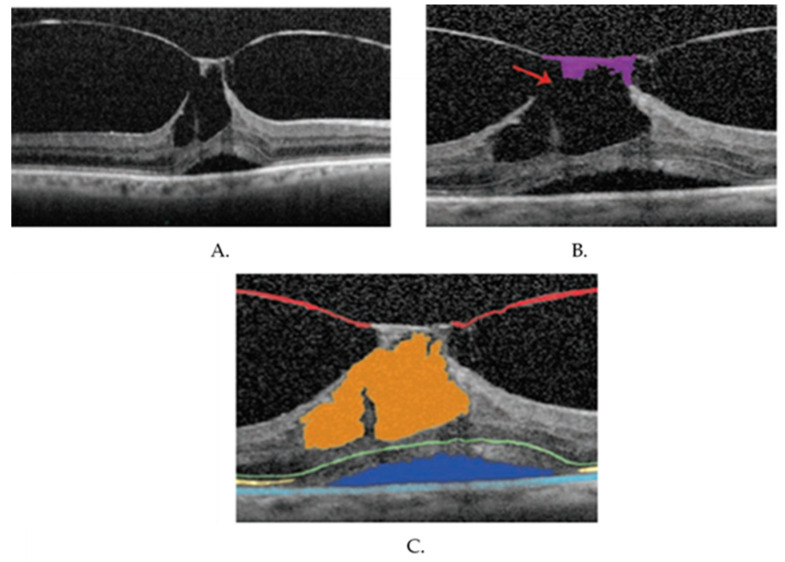
Extreme VMT causing foveal cysts: (**A**) VMT caused by partial PVD; (**B**) segmentation of roof of retina; (**C**) segmentation of PVD, subretinal fluid, and cyst.

**Figure 9 bioengineering-10-00407-f009:**
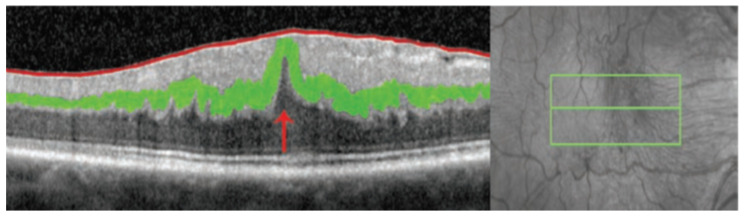
Retinal distortion due to the formation of ERM.

**Figure 10 bioengineering-10-00407-f010:**
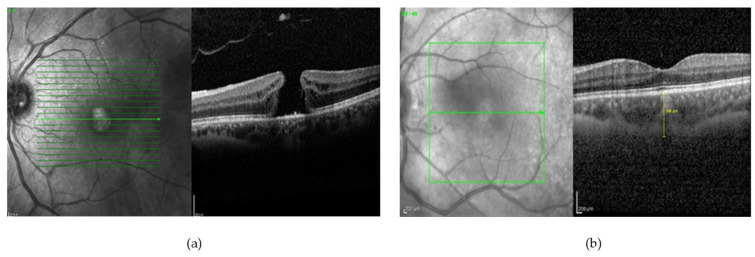
(**a**) Full-thickness hole; (**b**) lamellar macular hole.

**Figure 11 bioengineering-10-00407-f011:**
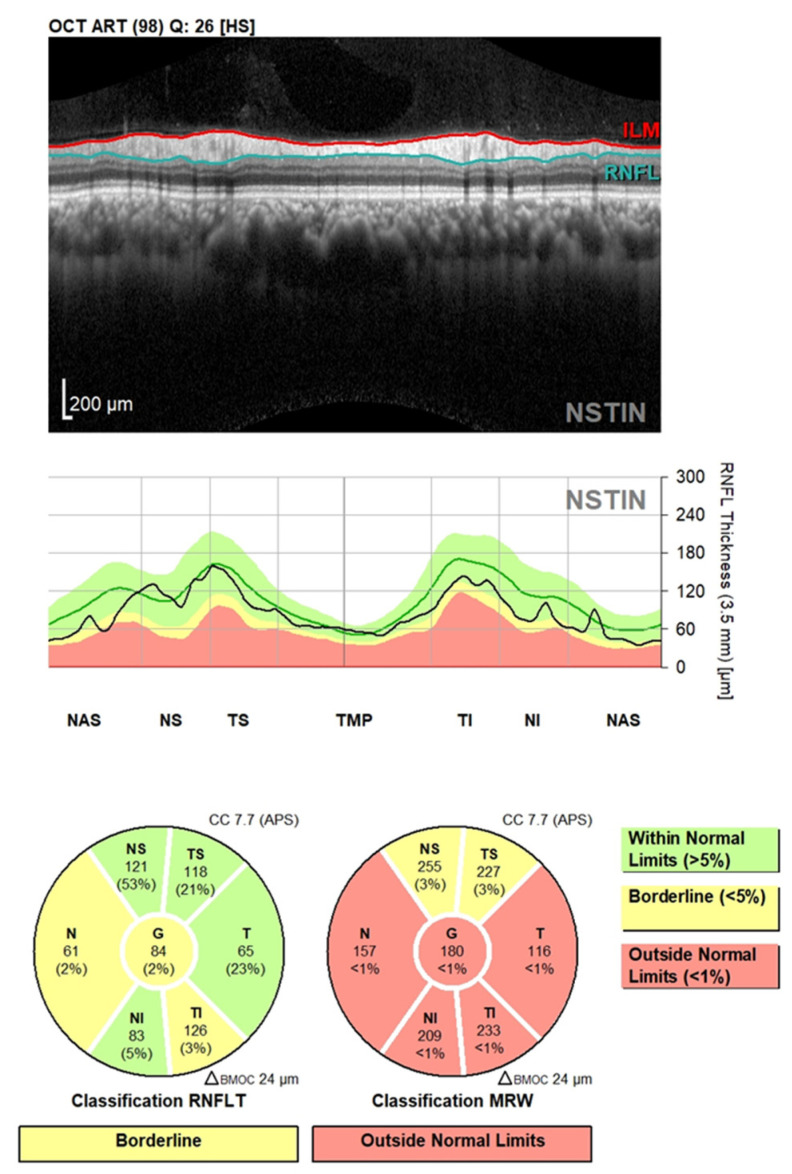
Glaucoma detection report based on RNFL thickness and MRW in various retinal zones using OCT.

**Figure 12 bioengineering-10-00407-f012:**
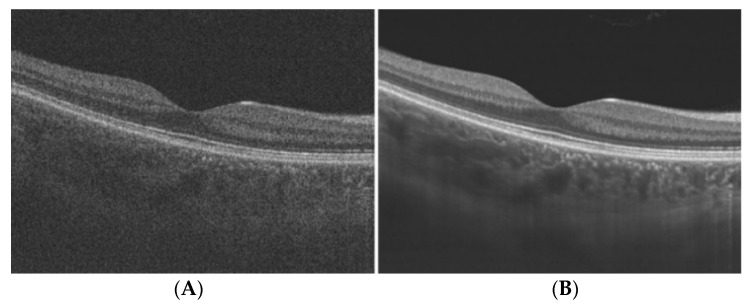
Speckle Noise reduction via super-resolution reconstruction: (**A**) noisy image; (**B**) denoised image.

**Figure 13 bioengineering-10-00407-f013:**
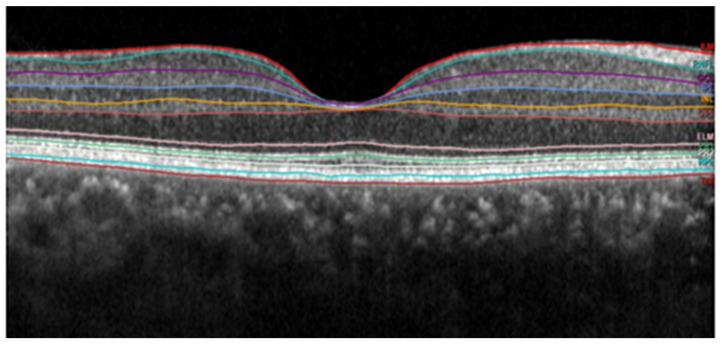
Subretinal layer segmentation of macular OCT.

**Figure 14 bioengineering-10-00407-f014:**
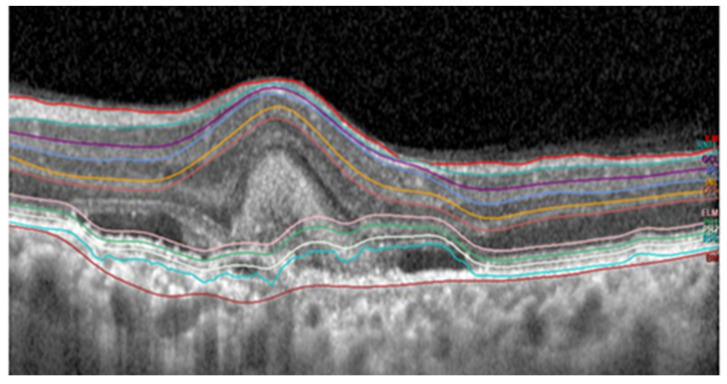
False segmentation of subretinal layers.

**Figure 15 bioengineering-10-00407-f015:**
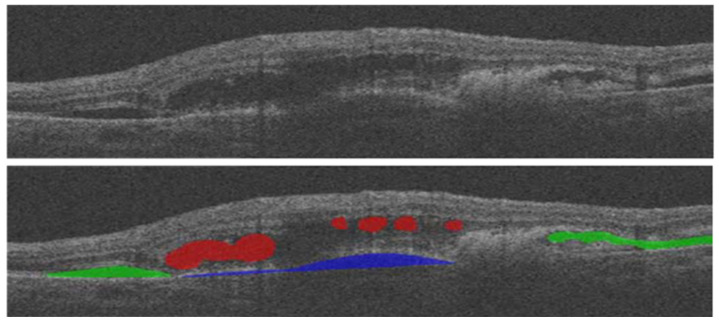
Various types of fluid detection shown in OCT.

**Figure 16 bioengineering-10-00407-f016:**
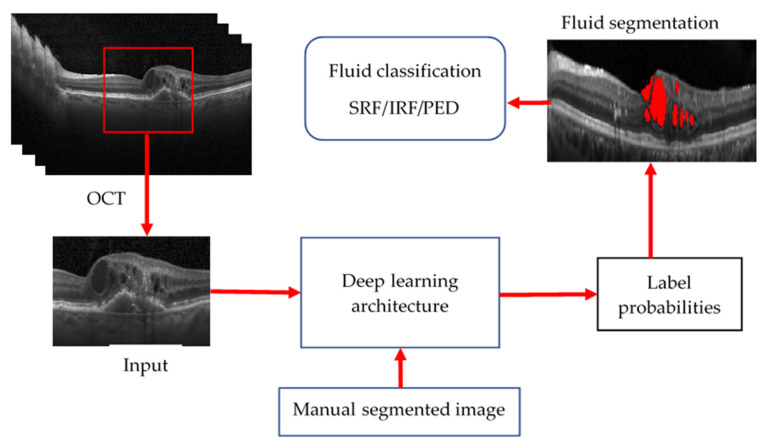
General procedure of AI-based OCT layer segmentation.

**Table 1 bioengineering-10-00407-t001:** Biomarker associated to eye disease.

Biomarkers	Association of the Biomarkers with the Eye Disease
Layer thickness	Changes in the retina’s thickness and its layers are characteristics of many diseases such as glaucoma and age-related macular degeneration (AMD). For example: a glaucoma patient has 20% lower RNFL (retinal nerve fibre layer) thickness than normal patient
Inner retinal lesion	A characteristic finding in various stages of diabetic macular oedema (DME) that is a key risk factor for developing more advanced stages of DME.
Drusen	A characteristic finding of the early stages of AMD that is a key risk factor for the development of more advanced stages.
Cup–disc ratio	A cup–disc ratio of more than 0.5 is a risk sign of glaucoma.
PVD	An early sign of macular 0edema and lamellar macular hole

**Table 2 bioengineering-10-00407-t002:** Pre-processing algorithms used for OCT segmentation.

Pre-Processing Methods	Researchers	Key Point	Evaluation Parameters
Deep learning (Unet and SRResNet and AC-SRResNet)	Y. Huang et al. [37]	This is the modification of the existing combination of U-Net and Super-Resolution Residual network with the addition of asymmetric convolution. The evaluation parameters used in this paper were signal-to-noise ratio, contrast to noise ratio, and edge preservation index.	SNR (dB) U-Net: 19.36 SRResNet: 20.11 AC-SRResNet: 22.15
SiameseGAN	K. Nilesh et al. [33]	This is the combination of Siamese network module and a generative adversarial network. This model helps generate denoised images closer to the ground-truth image.	Mean PSNR: 28.25 dB
Semi-Supervised (N2NSR-OCT)	Q. Bin et al. [31]	This paper utilises up- and down-sampling networks, consisting of modified U-net and DPBN, to obtain a super-resolution image.	PSNR: 20.7491 dB RMSE: 0.0956 dB MS: SSIM 0.8205
Semi-Supervised Capsule cGAN	M. Wang et al. [32]	This paper addresses the issue of speckle noise with a semi-supervised learning-based algorithm. A capsule cognitive generative adversarial network is used to construct the learning system, and the structural information loss is regained by using a semi-supervised loss function.	SNR: 59.01 dB
None	Yazdanpanah A. [38], Abramoff M.D. [39], Yang Q. [40] S. Bekalo et al. [41]	These researchers did not use any pre-processing algorithm for the oct image analysis.	DSC: 0.85 Correlation C/D: 0.93
2D linear smoothing	Huang Y. et al. [42]	This paper uses 3 × 3 pixel boxcar averaging filter to reduce speckle noise.	Avg. mean: 0.51 SD: 0.49
Mean filter	J. Xu et al. [14]	In this paper, the author investigated the possibility of variable-size super pixel analysis for early detection of glaucoma. The ncut algorithm is used to map variable-size superpixels on a 2D feature map by grouping similar neighbouring pixels.	AUC: 0.855
Wavelet shrinkage	Quellec G. et al. [43]	This paper describes an automated method for detection of the footprint of symptomatic exudate-associated derangements (SEADs) in SD-OCT scans from AMD patients. An adaptive wavelet transformation is used to extract a 3D textural feature.	AUC: 0.961
Adaptive vector-valued kernel function	Mishra A. et al. [44]	This paper proposes a two-step kernel-based optimisation scheme to identify the location of layers and then refine them to obtain their segmentation.	NA
Two 1D filters: (1) median filtering along the A-scans; (2) Gaussian kernel in the longitudinal direction	Baroni M. et al. [45]	The layer identification is made by smoothing the OCT image with a median filter on the A-scan and a Gaussian field on the B-scan image.	Correlation: 5.13 Entropy: 25.65
SVM approach	Fuller A.R. et al. [46]	This paper uses the SVM approach by considering a voxel’s mean value and variance with various resolution settings to handle the noise and decrease the classification error.	SD: 6.043
Low-pass filtering	Hee M.R. et al. [5]	The peak position of the OCT image was filtered out using a low-pass filter to create similarity in spatial frequency distribution in the axial position.	NA

**Table 3 bioengineering-10-00407-t003:** Overview of related work in fluid segmentation.

Reference	Fluid Type	Disease	Year	Method	Evaluation Parameter
Gopinath et al. [51]	IRF	AMD, RVO, DME	2019	Selective enhancement of cyst using generalised motion pattern (GMP) and CNN	Mean DC: 0.71
Y. Derradji et al. [52]	Retinal atrophy	AMD	2021	CNN and Residual U-shaped Network	Mean Dice score: 0.881 Sensitivity: 0.85 Precision: 0.92
Y. Guo et al. [53]	IRF, SRF, PED	DME	2020	ReF-Net	F1 score: 0.892
Marc Wilson et al. [54]	IRF, PED	AMD	2021	Various DL Models	DSC: 0.43–0.78
B. Sappa et al. [41]	IRF, SRF, PED	AMD	2021	RetFluidNet (based on auto-encoder)	Accuracy IRF: 80.05% PED: 92.74% PED: 95.53%
Girish et al. [55]	IRF	AMD, RVO, DME	2019	Fully connected neural network (FCNN)	Dice rate: 0.71
Venhuizen et al. [56]	IRF	AMD, RVO, DME	2018	Cascade of neural networks to form DL algorithm	DC: 0.75
Schlegl et al. [57]	IRF, SRF	AMD, RVO, DME	2018	Auto-encoder	AUC: 0.94
Retouch [1]	IRF, SRF, PED	AMD, RVO	2019	Various models proposed by participants	DSC: 0.7–0.8
Xu et al. [50]	Any	AMD	2015	Stratified sampling voxel classification for feature extraction and graph method for layer segmentation	TPR: 96% TNR: 0.16%
Chiu et al. [58]	Any	DME	2015	Kernel regression method to estimate fluid and graph theory and dynamic programming (GTDP) for boundary segmentation	DC: 0.78

## Data Availability

Data are contained within the article. The data discussed in this study are available as download links in Section 5.

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
