# Peer review of "On Machine Learning in Clinical Interpretation of Retinal Diseases Using OCT Images"

_bioengineering, 2023, doi:10.3390/bioengineering10040407_

Round 1

Reviewer 1 Report

This is a review of the state of knowledge about the field of image processing using OCT (Optical coherence tomography) for retinal disease. Although there are no new research results, the material is well-written and organized and should prove useful, especially so to students new to the field. Here are some points to consider. 1) References need to be cited from the reference list for most of the items mentioned in the introduction.  2) Figures 7 & 8 need the parts specified [as (a), (b)] in the title. 3) The title for Figure 12 is not complete. 4) The Data Availability says the links are available in Section 5. But some there do not have links which therefore should be added. 5) Most references have access data but the first one is missing. 

Author Response

Dear Reviewer,

We are writing this letter in response to your comments on our paper titled On Machine Learning in Clinical Interpretation of Retinal Diseases using OCT Images, which was recently submitted to Bioengineering for review. We thank you for reviewing our work and providing valuable feedback.

We appreciate your constructive comments and the effort you put into thoroughly reviewing our paper. Your comments have been fully addressed to improve the paper's quality.

Comments:

  • References need to be cited from the reference list for most of the items mentioned in the introduction.

Ans: Five references [1-4] & [8] have been added in the introduction section.

  • Figures 7 & 8 need the parts specified [as (a), (b)] in the title.

Ans: Figures 7 and 8 are now separated as (A, B, C) and explained in text.

  • The title for Figure 12 is not complete. 

Ans: The incomplete title of Figure 12 is now corrected.

  • The Data Availability says the links are available in Section 5. But some there do not have links which therefore should be added.

Ans: The download link of the missing dataset has been added now

  • Most references have access data but the first one is missing. 

Ans: Data have been added to the first reference and are now placed at number 5 in the reference list

We look forward to hearing from you regarding the revised manuscript.

Sincerely,

Prakash Kumar Karn

Waleed H. Abdulla

Reviewer 2 Report

The paper is well-organized and well-structured and also lacks some details for readers to understand. I would suggest the authors address the following concerns in order to meet the publication requirement:

1. The introduction section lacks fluidity and readability.

2.  A comparison with the state of the art in the form of a table should be added.

3. Authors should add the parameters of the algorithms.

4. Provide more visual representations of the results to help the reader understand and interpret the data, such as graphs or plots. The following papers are good examples:

https://doi.org/10.1016/j.eswa.2021.115406

https://doi.org/10.1155/2022/5052435

Author Response

Dear Reviewer,

We are writing this letter in response to your comments on our paper titled On Machine Learning in Clinical Interpretation of Retinal Diseases using OCT Images, which was recently submitted to Bioengineering for review. We thank you for reviewing our work and providing valuable feedback.

We appreciate your constructive comments and the effort you put into thoroughly reviewing our paper. Your comments have been fully addressed to improve the paper's quality.

Comments:

  • The introduction section lacks fluidity and readability.

Ans: The introduction section has been rewritten to maintain readability. It has now been divided into sub-sections to ease following up the content.

  • A comparison with the state of the art in the form of a table should be added.

Ans: Separate column has been added in table 2 and table 3 for the comparison of evaluation parameters and the key points in the state-of-art research.

  • Authors should add the parameters of the algorithms.

Ans: Evaluation parameters have been added in table 2 and table 3

  • Provide more visual representations of the results to help the reader understand and interpret the data, such as graphs or plots.

Ans: Figure 16 has been added to clarify the basic deep-learning algorithm used for OCT image analysis.

We look forward to hearing from you regarding the revised manuscript.

Sincerely,

Prakash Kumar Karn

Waleed H. Abdulla

Reviewer 3 Report

This review article was well written.

I have few suggestions to authors:

·        There was lot of basic material was presented in the OCT imaging and interpretation section and less was reviewed related to AI developments and use of AI developments in feature clinical OCT use.

·        Figure 12 legend was incomplete.

Author Response

Dear Reviewer,

We are writing this letter in response to your comments on our paper titled On Machine Learning in Clinical Interpretation of Retinal Diseases using OCT Images, which was recently submitted to Bioengineering for review. We thank you for reviewing our work and providing valuable feedback.

We appreciate your constructive comments and the effort you put into thoroughly reviewing our paper. Your comments have been fully addressed to improve the paper's quality.

Comments:

  • There was lot of basic material was presented in the OCT imaging and interpretation section and less was reviewed related to AI developments and use of AI developments in feature clinical OCT use.

Ans: Two paragraphs have been added to section 4.3 to illustrate the development of AI and its role in clinical OCT use.

  • Figure 12 legend was incomplete.

Ans: Figure 12 legend is now completed.

We look forward to hearing from you regarding the revised manuscript.

Sincerely,

Prakash Kumar Karn

Waleed H. Abdulla
